# The Medical, Clinical, and Radiographic Aspects of Multiple Idiopathic Tooth Resorption: A Systematic Review

**DOI:** 10.3390/jpm12071182

**Published:** 2022-07-20

**Authors:** Raphaël Richert, Julie Santamaria, Laurent Laforest, Jean-Christophe Maurin

**Affiliations:** 1Hospices Civils de Lyon, PAM d’Odontologie, 69007 Lyon, France; julie.santamaria@chu-lyon.fr (J.S.); jean-christophe.maurin@chu-lyon.fr (J.-C.M.); 2UFR d’Odontogie, Université Claude Bernard Lyon I, Université de Lyon, 69008 Lyon, France; laurent.laforest@chu-lyon.fr; 3Laboratoire de Mécanique des Contacts et Structures, UMR 5259 CNRS/INSA Lyon\Université de Lyon, 69100 Villeurbanne, France; 4Laboratoire des Multimatériaux et Interfaces, UMR CNRS 5615/UCBL, 69622 Villeurbanne, France; 5Laboratoire de Biologie Tissulaire et Ingénierie Thérapeutique, UMR 5305 CNRS\Université Claude Bernard Lyon 1, 69008 Lyon, France

**Keywords:** endodontics, tooth resorption, systematic review

## Abstract

Background: Many causes of resorption remain unclear and are thus identified as idiopathic. In such cases, management is difficult, especially when multiple teeth are involved. The aim of the present study was to assess the literature regarding the medical, clinical, and radiographic aspects of multiple idiopathic resorptions (MIR) and to examine the factors associated with the risk of extraction. Methods: The title and protocol were registered a priori in PROSPERO (CRD42020191564), and the study followed the PRISMA methodology. Four electronic databases were searched to include reviews and case reports on MIR in permanent dentition. Results: Among the 1035 articles identified, 31 case reports were included. The mean age of the patients was 32 years (SD = 16.4). MIR were consistently diagnosed after radiographic evaluation and were undetected during intra-oral examination in 62% of cases. The treatment involved extraction in 77% of cases. The risk of extraction increased in the presence of periodontal inflammation. Conclusions: MIR are aggressive forms of resorption requiring routine visits. MIR mostly involve extraction and lead to a challenging prosthetic rehabilitation due to severely damaged abutment teeth. However, the current knowledge on MIR remains fragmental and based on a limited number of case reports.

## 1. Introduction

According to the American Association of Endodontists, root resorption is a progressive process resulting in the loss of dentin and cementum [1]. This physiological process is essential for primary dentition, as it leads to the exfoliation of deciduous teeth. However, root resorption in permanent teeth is largely pathological and can be classified according to its location (internal/external, cervical/apical). Characteristics of external resorptions were widely analysed, and a three-stage mechanism was proposed to explain evolutions in vital teeth. First, damage at the cementum constitutes the portal of entry to initiate the resorption. Then, the resorption invaded the tooth structure, but the presence of a pericanalar resorption-resistant sheet protects the pulp from an early invasion. In the last stage, repair and remodelling phenomena took place with the apposition of bonelike tissue at different locations of the resorption [2]. Similar mechanisms were observed in the root canal-filled teeth, but the resorption stage was reported to be more intense as the pericanalar resorption-resistant sheet, already removed during the root canal treatment, cannot delay invasion [3]. In parallel, a number of potential predisposing factors have been identified, such as biomechanical forces following dental trauma, endodontic microorganisms and their toxins, developmental defects, neoplasia, and hormonal disturbances. All these causes result in an increased activity of osteoclastic cells and can lead to tooth avulsion if not detected and managed early [2,4,5].

Many causes of resorption are now known and have led to the adaptation of treatments, such as reducing the forces used in orthodontics or changing the protocols for bleaching [2,4,5]. The term idiopathic root resorption was first mentioned in 1930 by Mueller and Rony to define a case with no known aetiology [6]. Since this date, numerous clinical cases have been reported, and the definition has evolved due to a better understanding of the potential aetiologic factors, such as orthodontic reasons [7]. Nevertheless, clinical and aetiological aspects of some resorptions remain poorly understood, and the term idiopathic resorption is still used for numerous clinical cases. Such cases, particularly those concerning multiple idiopathic resorptions (MIR), are difficult to manage. Indeed, these aggressive forms progress rapidly, affect multiple teeth, and sometimes recur, despite intervention [8]. In MIR, communication with the patient is very difficult because no aetiology can be given to explain this pathology, which can even sometimes occur without symptomatology [9,10,11]. All these observations emphasise the need to gather the published data to increase the understanding of potential causes of MIR and adapt treatments. Hence, the aim of the present systematic review was to analyse the published data regarding the medical, clinical, and radiographic aspects of MIR and assess the potential factors associated with the risk of tooth extraction.

## 2. Materials and Methods

### 2.1. Protocol and Registration

The guidelines of the Preferred Reporting Items for Systematic Reviews and Meta-Analyses (PRISMA) [12] statement were followed to answer the following question: which medical, clinical, and radiographic aspects define multiple idiopathic tooth resorption. The methodology was previously registered in the International Prospective Register of Systematic Reviews (PROSPERO) database under the protocol number CRD42020191564.

### 2.2. Search Strategy

Three electronic databases were searched (PubMed/MedLine, Web of Science and the Cochrane Library). The search was performed as follows “multiple root resorption” OR “multiple tooth resorption” OR “numerous root resorption” OR “numerous tooth resorption” OR “multiple teeth resorption” OR “numerous teeth resorption” OR “idiopathic root resorption” OR “idiopathic tooth resorption” OR “idiopathic teeth resorption” (Table 1). The last search was performed on 9 January 2022, with no limit regarding the year of publication. The records identified were imported from each database and saved into a spreadsheet (Excel Office 360, Microsoft, Redmond, WA, USA).

### 2.3. Eligibility Criteria

The eligibility criteria for inclusion were: cohort, case-control, cross-sectional studies, and case reports of multiple idiopathic root resorptions on permanent dentition, presence of medical, clinical, or radiographic data in the full-text article, and full text available in English. The exclusion criteria were: unique internal or external resorption, presence of an orthodontic or whitening treatment, and antecedent of trauma.

### 2.4. Study Selection

The titles and abstracts were screened independently by two reviewers. The full texts of all the abstracts in accordance with the inclusion criteria were collected and reviewed (by consensus). In addition, the bibliographies of the considered papers were scanned to identify additional missing relevant articles. Again, a consensus between the two reviewers was reached to determine which studies met the inclusion criteria. In case of conflict, a third reviewer was consulted.

### 2.5. Quality of Evidence Assessment

The quality of the included articles was evaluated using the Case Report (CARE) and the Preferred Reporting Items for Case Reports in Endodontics (PRICE) guidelines [12]. The checklists are available in the Appendix A. After applying the checklists, a mean compliance per item was calculated for all the articles included.

### 2.6. Data Extraction and Statistical Analysis

Among the different studies identified, patient characteristics including the potential medical, clinical, and radiographic correlates of the risk of extraction were collected. Data were recorded in an EXCEL file using the following binary factors: age (+/– 30 years), sex, symptomatology (y/n), gingival and dental health (y/n), pulp vitality (y/n), tooth mobility (y/n), x-ray detection based on a cone beam computed tomography (CBCT) (y/n), number of teeth affected (+/– 10 teeth), location of the resorption (anterior/posterior, external/internal, cervical/apical), type of treatment (extraction, restoration, both, refusal, control visit), further resorption (y/n). For this analysis, the treatment variable was dichotomized into at least one extraction (possibly associated with restorations) vs. no extraction. The statistical significance threshold level was set at 0.05. When the conditions of validity of the Chi-squared test were not met, Fisher’s exact test was used.

## 3. Results

### 3.1. Search Results

The electronic search identified a total of 1035 articles, after the removal of duplicates. The screening of the abstracts led to exclude 965 records, while 70 were found to be eligible for full-text screening. After reading the 70 articles selected, 39 were excluded for reasons listed in Figure 1. The remaining 31 studies were screened for data on medical, clinical, and radiographic factors [5,7,8,9,10,11,13,14,15,16,17,18,19,20,21,22,23,24,25,26,27,28,29,30,31,32,33,34,35,36,37,38].

### 3.2. Quality of Evidence Assessment

The included studies described only case reports, three of which were associated with systematic reviews [7,23,24]. The detailed results of the quality of evidence assessment using the CARE and PRICE guidelines are presented in the Appendix A (Appendix A, respectively). The mean compliance was 67% with a maximum score of 100% and a minimum score of 6% for the item “extra-oral findings”. The item “inter-vention adherence” was fulfilled in only 9% of the reports.

### 3.3. Formatting of Mathematical Components

Among the 31 studies included, 4 of them described several (2 or 3) case reports, among which 2 of them included patients with a familial pattern (i.e., 2 patients per case report) [8,15]. Overall, 37 clinical cases were investigated. The mean age of patients was 32 years old, 41% of patients were older than 30 years, and 54% were males. The MIR was asymptomatic in 38% of cases and was not detected during the intraoral examination in 62% of cases. Among the 22 cases for which thermal sensitivity was mentioned, the latter was positive in 86% of cases. A periodontal inflammation was present in 37% of cases but only evaluated in 27 cases. A radiographic evaluation was systematically performed to establish the diagnosis. The MIR was classified as external in 84% of cases, cervical in 68% of cases, and affected the four quadrants in 76% of cases. In 50% of cases, the number of affected teeth was superior to 10 (Table 2).

All parameters were considered as binary factors: 1 was considered when at least one extraction was conducted and 0 in other cases; 1 was considered when at least one cervical location was present and 0 in other cases.

Among the 37 clinical cases reported, 5 underwent only restorations, 27 presented at least one extraction (possibly associated with restorations), 2 refused treatment, and 3 only underwent control visits. Among the 35 patients who did not refuse treatment, at least one extraction was performed in 77% of cases. Further resorption occurred in 55% of cases but was only evaluated in 27 cases. A tooth extraction was performed in all patients aged over 30 years old compared to 62% of patients under 30 years old (*p* = 0.01). Likewise, all patients presenting a periodontal inflammation had a tooth extraction, while only 53% of patients without a periodontal inflammation underwent an extraction (*p* = 0.02). A tooth extraction was performed significantly more for a cervical location of the MIR compared to apical ones (92% vs. 45%, respectively, *p* = 0.005). Although not statistically significant, the extraction of at least one tooth was higher when MIR was undetected at the intraoral examination (79% vs. 64%, *p* = 0.41), in the presence of symptomatology (82% vs. 69%, *p* = 0.43), and when CBCT was used (83% vs. 76%, *p* = 1).

## 4. Discussion

The aim of this systematic review was to report the medical, clinical, and radiographic aspects of MIR. The present review revealed that the current knowledge on MIR is mostly based on studies of low-level evidence and limited data. This review also found that MIR is difficult to diagnose without any radiographic evaluation. Most treatments led to extractions without any guarantee that the resorption process, at the origin of further lesions, would stop.

Several limitations concerning the current analysis should be discussed first. Despite the high number of screened articles, only a minority presented the selected inclusion criteria. This observation should lead the reader to consider the present results with caution, given the low counts available for statistical analysis. This, however, reflects the paucity of the literature in this field. Moreover, given that the available studies were only case reports, high-level evidence concerning the different aspects of MIR is missing. Besides, MIR was defined herein as multiple root resorptions which develop in the absence of a plausible cause. Consequently, idiopathic root resorption is a diagnosis of exclusion and multifactorial [7]. However, it is important to note that what is considered idiopathic has evolved over time [39]. Nowadays, multiple resorptions are often reported to be due to orthodontics or trauma, but this reason of exclusion was never mentioned in the included studies. Moreover, some recently discovered causes of resorption, such as Paget’s disease or virus cat infection [40,41], were not likely to have been investigated in the older studies, which concluded with idiopathic root resorptions. However, the present review excluded all studies that mentioned these newly discovered causes of resorption. Two case reports involving a familial pattern were included herein but might be considered non-idiopathic in the future if the exact genetic causes are discovered. Furthermore, the risk of bias was high, as different decisive items of the CARE and PRICE checklists related to follow-up or diagnostic testing were not always fulfilled.

The reported epidemiological factors related to MIR are unclear and sometimes controversial. Although the present results found a similar prevalence between males and females [9,10], a higher prevalence for males was previously reported [11]. Similarly, MIR was reported to occur more frequently in younger patients [28,30], but opposite findings were also observed [7,24]. Although the mean age defined herein was 32 years old, the age ranged largely from 14 to 93 years. All these discordances could be explained by the small number of clinical cases reported in the literature, further emphasising the need to remain cautious when comparing findings between studies.

MIR frequently affects a large number of teeth, which are more likely to be extracted if the MIR is undetected. These observations confirm the very aggressive form of this pathology and emphasize the need for early detection [8]. However, the diagnosis of MIR can be complex, especially at the early stages of the pathology, since these resorptions are often undetected during routine examinations [9,10,11]. Thermal tests often respond within normal limits but could be negative depending on the extent of the lesion. Moreover, the pink spot is not always present [8,9]. In these conditions, periodontal probing could be useful to diagnose the presence of MIR, although this procedure was mentioned in only a few clinical cases. All these observations lead to the fact that radiographic evaluation is essential in MIR detection [37,38]. More recently, the use of CBCT has been frequently reported to manage MIR [22,28]. However, the benefit of CBCT alone is difficult to assess considering the recent developments in terms of conservative approaches and biomaterials used [4]. In cases of doubt, CBCT still constitutes the key complementary test to detect early forms of resorption or even control their progression without any intervention [29].

The present review suggests that MIR could be associated with periodontal inflammation, but the exact relation with periodontal disease remains unclear [7]. Moreover, the risk of extraction was also associated with the presence of periodontal inflammation and the cervical location of MIR. When the resorption is cervical, it enables periodontal inflammation and early evolution to extensive lesions [39]. The decision between a conservative approach or extraction is traditionally based on accessibility and extension of the lesion [42]. For MIR, this decision-making process could lead to consider frequent extractions and prematurely complete removable dental prostheses [26,34]. However, the difficulties raised by cervical lesions are not restricted to MIR, and this location constitutes a high-risk factor of failure for treatment of carious and non-carious lesions [43]. The management of subgingival carious lesions faces difficulties of coronal leakage and recurrent cavities, but also, cervical lesions concentrate higher mechanical stress in the dentinal walls [44]. Recently, silicate-based materials have changed the way to treat perforations and cervical lesions, which is an opportunity for a more conservative management of MIR.

A surgical repair without root canal treatment was indeed considered to treat MIR using mineral trioxide aggregate (MTA) and could be recommended for superficial lesions. This surgical approach could also be combined with a root canal treatment for more extensive lesions [10,35]. Non-idiopathic cavities, which communicate with the oral cavity, were managed by Patel et al. using composite resin or glass ionomer cement restoration for their good mechanical and aesthetic properties or silicate-based materials for their proven biocompatibility and sealing ability [42]. However, it is of particular importance to note that this choice of a direct restoration is motivated by the fact that only one tooth is affected and did not support any removable prosthesis. Considering MIR, all quadrants were affected, leading to a complex prosthetic rehabilitation based on fixed and/or removable prostheses. Therefore, the tissue loss of each restored or abutment tooth and the occlusal scheme should be cautiously analysed to better transmit forces and avoid early vertical root fracture. If most teeth are at risk of fracture, implant-supported complete prostheses represent a valuable solution in the long term to preserve healthy bone support [26,34]. However, these approaches also raise socio-economic challenges, as the cost of implants might appear adapted to a minority. This difficult care access could even destroy the trust in the practitioner, as the patient might even be asymptomatic, and etiologic reasons remain unclear. In some cases, this could even lead to a refusal, as reported by Celikten et al. [45].

Decision making between a conservative approach or extraction is even more complicated by the fact that further resorption was sometimes found after an initial conservative treatment. A possible explanation could be that small resorptions might go undetected during the radiographic evaluation and evolve after treatment [38]. However, it was found that resorption continued even in the MTA-treated teeth [9,35]. It was also reported that MIR can recur because of the aggressive nature of the osteoclastic activity [8], but resorption might also be influenced by different growth or sex hormones [27,46] and environmental factors such as changes in the oral flora or dietary chemicals [21]. To date, the list of the suspected, but not yet proven, factors of resorption include: systemic conditions and syndromes (e.g., hypothyroidism, hyperparathyroidism, systemic sclerosis, Gaucher’s disease, hereditary haemorrhagic telangiectasia, Paget’s disease, Goltz syndrome, Papillon–Lefévre syndrome, and Turner syndrome), altered osteoclast activity due to genetic mutations (e.g., familial expansile osteolysis), medication-induced resorption (e.g., bisphosphonates), and viral infections (e.g., herpes zoster) [41]. The point is that to ‘render proper treatment’, root resorption invariably relies on ‘removing the etiological factor’ [47] which, in the case of MIR, remains unknown. In the absence of an understanding of the pathological processes underlying a MIR, regular clinical and radiographic monitoring seems to be decisive to preserve the teeth and prevent further root resorption.

## 5. Conclusions

The present review demonstrated that MIR represents a very aggressive form of resorption that can frequently affect more than 10 teeth and is mostly treated by extraction. The intra-oral examination does not allow a systematic detection of the resorption, emphasizing the need for routine clinical visits. In cases of doubt, CBCT could constitute a key complementary test to detect early forms of resorption.

A stagged approach and early management appear recommended considering the high number of affected teeth. For small lesions, a surgical approach and vital pulp therapy should be first attempted. For more extensive lesions, root canal treatment with composite resin or glass ionomer cement restoration will probably maintain teeth for many years. At the last stage, and if most teeth are at risk of crown fracture, progressive extraction and implant-supported prostheses will become necessary.

However, the present review reveals that the current knowledge on MIR remains fragmental and based on a limited number of case reports, with often limited data. This emphasises the need to more systematically report idiopathic clinical cases following the CARE and PRICE statements to more effectively assess the potential factors of resorption.

## Figures and Tables

**Figure 1 jpm-12-01182-f001:**
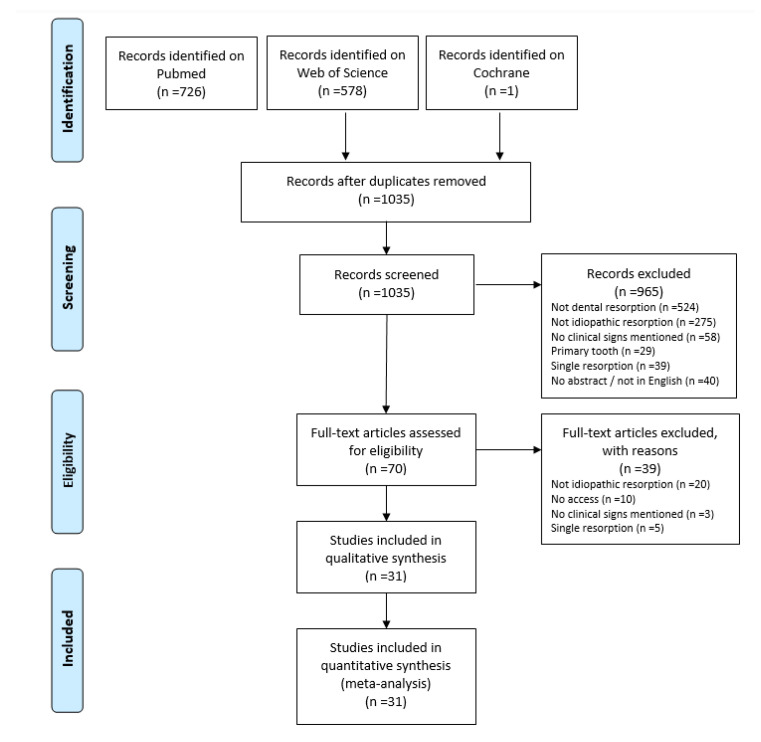
Flow diagram of the screening and selection process adapted from the Preferred Reporting Items for Systematic Reviews and Meta-Analyses (PRISMA) statement [11].

**Table 1 jpm-12-01182-t001:** Electronic database and search strategy (9 January 2022).

Database	Search
MEDLINE [PubMed]	“multiple root resorption” OR “multiple tooth resorption” OR “numerous root resorption” OR “numerous tooth resorption” OR “multiple teeth resorption” OR “numerous teeth resorption” OR “idiopathic root resorption” OR “idiopathic tooth resorption” OR “idiopathic teeth resorption”.
Web of Science	“TITLE–ABS -KEY + multiple + AND + root + AND + resorption + OR + TITLE–ABS–KEY + multiple +AND + tooth + AND resorption + OR + TITLE–ABS–KEY + numerous +AND + tooth + AND resorption + OR + TITLE–ABS–KEY + numerous +AND + root + AND resorption + OR + TITLE–ABS–KEY + multiple +AND + teeth + AND resorption + OR + TITLE–ABS–KEY + numerous +AND + teeth + AND resorption + OR + TITLE––ABS–KEY + idiopathic +AND + tooth + AND resorption + OR + TITLE–ABS–KEY + idiopathic +AND + teeth + AND resorption+ OR + TITLE–ABS–KEY + idiopathic +AND + root + AND resorption”
Cochrane Library	“TITLE–ABS -KEY + multiple + AND + root + AND + resorption + OR + TITLE––ABS–KEY + multiple +AND + tooth + AND resorption + OR + TITLE–ABS–KEY + numerous +AND + tooth + AND resorption + OR + TITLE–ABS–KEY + numerous +AND + root + AND resorption + OR + TITLE–ABS–KEY + multiple +AND + teeth + AND resorption + OR + TITLE–ABS–KEY + numerous +AND + teeth + AND resorption + OR + TITLE–ABS–KEY + idiopathic +AND + tooth + AND resorption + OR + TITLE–ABS–KEY + idiopathic +AND + teeth + AND resorption+ OR + TITLE––ABS–KEY + idiopathic +AND + root + AND resorption”

**Table 2 jpm-12-01182-t002:** Main characteristics and parameters of the included studies.

Study	Age	Sex	Symptoms	Intra-Oral Detection	Periodontal Inflammation	Thermal Response	X-ray Detection	Number of Teeth Affected	Location	Treatment	Further Resorption
**Chen et al. 2020** [38]	29	woman	no	yes (and pink spot)	yes	yes	3d	29	ant and post internal cervical	extraction and restoration (MTA)	yes
**Sharma et al. 2019** [9]	23	woman	yes	no	no	yes	3d	16	ant and post internal cervical	extraction and restoration (MTA, RPD)	yes
**Abdullah et al. 2017** [16]	24	man	no	yes	no	yes	3d	2	ant external cervical and apical	restoration (MTA)	no
**Neely et al. 2016** [8]	93	man	no	not evaluated	not evaluated	not evaluated	2d	11	ant and post, external cervical and apical	extraction and restoration (N.A)	yes
**Neely et al. 2016** [8]	60	woman	no	yes (and pink spot)	no	not evaluated	2d	2	ant internal cervical	extraction and restoration (ISP)	yes
**Choudhury et al. 2015** [37]	28	man	yes	yes	yes	yes	2d	18	ant and post internal and external cervical and apical	extraction and restoration (RPD)	not evaluated
**Bansal et al. 2015** [11]	22	man	yes	yes	not evaluated	yes	2d	32	ant and post external apical	Restoration (N.A)	not evaluated
**Celikten et al. 2014** [36]	36	man	yes	no	no	yes	3d	11	ant and post internal and external cervical	refusal	not evaluated
**Jiang et al. 2014** [10]	26	woman	yes	not evaluated	not evaluated	not evaluated	2d	7	ant external cervical	extraction and restoration (GIC, MTA)	yes
**Kalender et al. 2014** [35]	33	man	yes	no	no	yes	3d	18	ant and post internal cervical	extraction (N.A)	yes
**Haeberle et al. 2013** [34]	19	woman	no	yes	yes	yes	2d	not evaluated	ant and post external cervical	extraction (ISP)	all teeth extracted
**Kanungo et al. 2013** [33]	16	man	yes	yes (and pink spot)	no	no	2d	3	post external apical	control	not evaluated
**Roy et al. 2012** [32]	32	man	yes	not evaluated	not evaluated	yes	2d	10	ant and post external cervical	extraction and restoration (RPD)	yes
**Arora et al. 2012** [31]	36	woman	yes	no	yes	no	3d	7	ant and post external cervical	extraction and restoration (N.A)	yes
**Yu et al. 2011** [30]	33	man	yes	no	no	yes	3d	32	ant and post external cervical	extraction (RPD)	yes
**Khojastepour et al. 2010** [29]	17	man	no	no	no	yes	2d	8	ant and post external apical	control	not evaluated
**Mattar et al. 2008** [27]	23	woman	yes	no	no	yes	2d	8	ant and post external cervical	both (N.A)	no
**Gupta et al. 2008** [26]	38	woman	no	not evaluated	not evaluated	yes	2d	14	ant and post external apical	extraction (RPD)	not evaluated
**Nikolidakis et al. 2008** [25]	46	woman	yes	yes	yes	yes	2d	4	ant external cervical	extraction and restoration (GIC)	no
**Sogur et al. 2008** [23]	18	woman	yes	no	no	yes	2d	32	ant and post external apical	extraction and restoration (RC)	no
**Iwamatsu-Kobayashi et al. 2005** [24]	49	woman	yes	no	no	yes	2d	21	ant and post external cervical	extraction and restoration (GIC, RPD)	no
**Liang et al. 2003** [7]	19	woman	yes	no	no	not evaluated	2d	12	ant and post external cervical	extraction and restoration (N.A)	yes
**Liang et al. 2003** [7]	68	man	yes	no	not evaluated	not evaluated	2d	7	ant and post external cervical	extraction (N.A)	yes
**Liang et al. 2003** [7]	50	man	no	no	not evaluated	not evaluated	2d	11	ant and post external cervical	extraction (N.A)	yes
**Domizio et al. 2000** [22]	26	woman	yes	not evaluated	not explained	not evaluated	2d	9	ant and post external apical	extraction (N.A)	yes
**Lynch and Ahlberg 1994** [14]	28	man	yes	yes	no	yes	2d	2	post external cervical	restoration (Amg)	no
**Rivera and Walton 1994** [21]	24	man	no	no	no	yes	2d	15	ant and post external apical	control	no
**Moody and Muir 1991** [20]	19	woman	no	no	yes	not evaluated	2d	6	ant and post external cervical	extraction and restoration (RPD)	no
**Moody et al. 1990** [19]	27	man	yes	no	yes	not evaluated	2d	16	ant and post external cervical	extraction and restoration (Amg)	yes
**Moody et al. 1990** [19]	20	man	yes	no	not evaluated	not evaluated	2d	9	ant and post external cervical	extraction (N.A)	yes
**Moody et al. 1990** [19]	44	woman	yes	no	not evaluated	not evaluated	2d	7	ant and post external cervical	Extraction (RPD)	not evaluated
**Yusof and Ghalazi 1989** [18]	35	man	yes	yes	yes	no	2d	5	ant and post external apical	extraction and restoration (N.A)	not evaluated
**Lydiatt et al. 1989** [17]	39	man	no	yes	yes	not evaluated	2d	14	ant and post external cervical	extraction (RPD)	no
**Pankhurst 1989** [13]	14	man	no	no	no	not evaluated	2d	20	ant and post external apical	refusal	not evaluated
**Saravia and Meyer 1989** [15]	14	woman	no	no	no	not evaluated	2d	4	post external apical	restoration (N.A)	no
**Saravia and Meyer 1989** [15]	14	woman	yes	no	no	not evaluated	2d	8	post external apical	restoration (N.A)	no
**Pankgurst et al. 1988** [13]	30	man	no	no	yes	yes	2d	14	ant and post external apical	extraction and restoration (N.A)	no

Note 2d: two dimensional; 3d: three dimensional; ant: anterior; post: posterior; MTA: mineral trioxide aggregate; RPD: removable partial denture; N.A: not available; ISP: implant-supported prosthesis; GIC: glass ionomer cement; RC: resin composite; Amg: amalgam.

## Data Availability

The data presented in this study are available on request from the corresponding author.

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
