# Peer review of "The Medical, Clinical, and Radiographic Aspects of Multiple Idiopathic Tooth Resorption: A Systematic Review"

_jpm, 2022, doi:10.3390/jpm12071182_

Round 1
Reviewer 1 Report
Dear Authors
The manuscript is well written , organised and with consistent data about the subject, and may be eligible to be published.
Thank you
Author Response
We would like to thank the reviewer for these acknowledgments.

Reviewer 2 Report
Comments and suggestions:
The topic of the present systematic review, discussing the medical, clinical, and radiographic aspects of multiple idiopathic tooth resorption, is interesting and undercovered in the literature.
The study protocol is adequate to the requisites for a systematic review.
The manuscript seems well structured.
The issue has been adequately described in the Introduction section and comprehensively discussed.
Reviewed findings currently presented may pave the way for further clinical investigations and may be clinically relevant in the future in the perspective of management of this disorder. Therefore, I would suggest to add a paragraph synthesizing from a clinical point of view providing evidence-based recommendations for multiple idiopathic tooth resorption management to clinicians.
Author Response
Our response: We agree with this comment and would like to thank the referee for this constructive feedback. Strategies for management of multiple resorptions were presented in the Discussion section of the initially submitted version of the manuscript. A paragraph synthesizing clinical recommendations has been added in the Conclusion section of the revised manuscript.
Revised text:
In Discussion section
A surgical repair without root canal treatment was indeed considered to treat MIR using Mineral Trioxide Aggregate (MTA) and could be recommended for superficial lesions. This surgical approach could also be combined with a root canal treatment for more extensive lesions [10,35]. Non-idiopathic cavities, which communicate with the oral cavity, were managed by Patel et al. using composite resin or glass ionomer cement restoration for to their good mechanical and aesthetic properties or silicate-based materials for to their proven biocompatibility and sealing ability [42]. However, it is of particular importance to note that this choice for a direct restoration is motivated by the fact that only one tooth is affected and did not support any removable prosthesis. Considering MIR, all quadrants were affected leading to a complex prosthetic rehabilitation based on fixed and/or removable prostheses. Therefore, the tissue loss of each restored or abutment tooth and the occlusal scheme should be cautiously analyzed to better transmit forces and avoid early vertical root fracture. If most teeth are at risk of fracture, implant-supported complete prostheses represent a valuable solution at long term to preserve healthy bone support [26,34]. However, these approaches also raise socio-economic challenges, as the cost of implants might appear adapted to a minority. This difficult care access could even destroy the trust in the practitioner as patient might even be asymptomatic, and etiologic reasons remain unclear. In some cases, this could even lead to refusal as reported by Celikten et al [45].
In conclusion section
A stagged approach and early management appear recommended considering the high number of affected teeth. For small lesions, a surgical approach and vital pulp therapy could be firstly attempted. For more extensive lesions, root canal treatment with composite resin or glass ionomer cement restoration will probably maintain teeth for many years. At a last stage and if most teeth are at risk of crown fracture, progressive extraction and implant-supported prostheses will become necessary.
